# The Immunoprotective Effect of ROP27 Protein of *Eimeria tenella*

**DOI:** 10.3390/ani13223500

**Published:** 2023-11-13

**Authors:** Menggang Li, Xiaoling Lv, Mingxue Zheng, Yingyi Wei

**Affiliations:** 1Guangxi Key Laboratory of Animal Breeding, Disease Control and Prevention, College of Animal Science and Technology, Nanning 530004, China; limenggang2023@163.com; 2College of Veterinary Medicine, Shanxi Agricultural University, Jinzhong 030801, China; lvxiaoling19@163.com (X.L.); zhengmingxue288@163.com (M.Z.); 3Guangxi Zhuang Autonomous Region Engineering Research Center of Veterinary Biologics, Nanning 530004, China

**Keywords:** *Eimeria tenella*, recombinant *Et*ROP27 protein, expression changes, immunoprotective

## Abstract

**Simple Summary:**

*Eimeria tenella* is the most pathogenic and common coccidia that causes chicken coccidiosis. At present, the control of this disease mainly relies on anticoccidial drugs, but the use of these drugs is greatly limited due to issues such as drug-resistant strains and drug residues. This study aims to investigate the immunoprotective effect of recombinant *Et*ROP27 virulence protein, which can provide theoretical support for the development of chicken coccidiosis vaccines. This study successfully constructed a prokaryotic expression vector of *Et*ROP27 and purified the recombinant protein, identified the expression pattern of the protein, and tested its immune protection. *Et*ROP27 has been shown to have a certain immunoprotective effect.

**Abstract:**

*Eimeria tenella* rhoptry protein has the properties of a protective antigen. *Et*ROP27 is a pathogenic gene that is detected via a transcriptome, but its expression pattern, immunogenicity, and potency are unknown. Therefore, a gene segment of *Et*ROP27 was amplified and transplanted into the pET28a prokaryotic vector for the expression of the recombinant protein, and it subsequently purified for the generation of a polyclonal antibody. Then, RT-PCR and Western blotting were performed to understand the expression pattern of *Et*ROP27. Subsequently, animal experiments were conducted to evaluate the immunoprotective effect of the recombinant protein with different immunizing doses (50, 100, and 150 μg). The results showed that the expression of *Et*ROP27 gradually increased with the prolongation of infection time, reaching the highest level at 96 h and then decreasing. Additionally, *Et*ROP27 is a natural antigen of coccidia that can stimulate the body to produce high levels of IgY. As with recombinant protein vaccines, the results of immune protection evaluation tests showed that the average weight gain rates of the immune challenge groups were significantly higher than that of the challenged control group, and their average lesion scores were significantly lower than that of the challenged control group. Furthermore, the oocyst excretion decreased by 81.25%, 86.21%, and 80.01%, and the anticoccidial index was 159.45, 171.47, and 166.75, respectively, for these groups. *Et*ROP27 is a promising antigen gene candidate for the development of a coccidiosis vaccine.

## 1. Introduction

Chicken coccidiosis is a parasitic protozoonosis caused by one or more of the seven species of *Eimeria* that are parasitic within the epithelial cells of the chicken intestinal tract [1,2]. The economic losses caused by chicken coccidiosis infection worldwide amount to GBP 10.4 billion annually [3]. China spends hundreds of millions of CNY annually on medicines to prevent and control coccidiosis, accounting for one-third of the cost of chicken disease prevention and control, and it is the second largest chicken production and consumption country in the world [4]. Coccidiosis is characterized by intestinal injury, diarrhea, or bloody stools [5]. *Eimeria tenella* is the most virulent, widely distributed, and harmful of the seven chicken coccidia species, mainly attacking the cecum [6]. The highest incidence rate was found in 3~6-week-old chicks. The symptoms of excreting bloody stool or even blood only appear after 4~5 days of infection. Large numbers of deaths begin one to two days after the appearance of bloody stools, with a mortality rate of up to 80% in severe cases [7,8,9].

Drug control has been the main method for chicken coccidiosis for a long time. However, many problems such as the emergence of drug-resistant strains, drug residues, and increased treatment costs during the prevention and treatment process have greatly limited drug control [3]. Therefore, the key strategy and methods for effective prevention and control of this disease is to develop a safe, effective, low-toxicity, and environmentally friendly vaccine against coccidiosis in chickens. Currently, live coccidian oocyst vaccines, with virulent and precocious strains, are widely used. However, traditional coccidiosis vaccines have shortcomings that limit their wide application such as high production costs, cumbersome production processes, environmental pollution caused by the spread of pathogens to the outside world, and the improper use of wild virus live worm vaccines that can cause outbreaks of coccidiosis [10]. Recent studies have demonstrated that recombinant protein vaccines can be used as effective control measures against coccidiosis [11].

Rhoptry proteins (ROPs) are conserved and immunoprotective in nature, and some members of this protein family are among the main virulence factors of *Toxoplasma gondii* [12]. Therefore, this protein family is considered as a potential target in the development of anti-*toxoplasmosis* vaccines. As one of the key virulence factors of the *T. gondii* type I strain, ROP18 participates in multiple mechanisms to escape host immune response. ROP5 controls the activity of ROP18 and affects IFN-γ and other effectors dependent on IRG protein, Irgm3, to regulate the acute toxicity response of *T. gondii* [13]. Similarly, ROPs play an important role in the process of *E. tenella* invading host cells. They exhibit the property of a protective antigen, and their corresponding antibodies can inhibit the infection of the host by coccidia [14]. Research has proven that *Et*ROP17 can inhibit host cell apoptosis [15]. In ROPs such as ROP14, ROP30, ROP52, and ROP53, *E. tenella* was detected in the proteome at different developmental stages; they have some degree of homology with the ROPs of *T. gondii* [15], but their functions are unclear. Zhang Li used RNA-Seq to detect and compare the transcriptome of the precocious strain and the virulent strain at the mitotic stage. qPCR was used to verify and screen the pseudopathogenic gene ROP27 of *E. tenella*, which is presumed to have a good immunogenicity [16,17]. However, it is not yet known whether the recombinant protein can effectively prevent the *E. tenella* disease.

Therefore, this study aims to investigate whether chicken *Et*ROP27 can reduce the effects of *E. tenella* infection by expressing the recombinant protein ROP27 and detecting its immunoprotective effect. Additionally, theses results may influence the development of an effective coccidiosis vaccine that can be used on a commercial scale.

## 2. Materials and Methods

### 2.1. Ethics Statement

All experiments involving animals were carried out in accordance with national regulations for the protection of animal welfare, and these guidelines were strictly followed. Additionally, this study was approved by the Animal Experimental Ethical Inspection Form of Guangxi University, China (Approval Code: Gxu-2023-0197; Approval Date: 16 February 2023). This study did not involve any human research.

### 2.2. Experimental Animals and Parasite

Firstly, 15-day-old SPF chicken embryos were obtained from Beijing Meri Avigon Laboratory Animal Technology Co., Ltd. (Beijing, China). Secondly, the *E. tenella* Shanxi virulent and precocious strains were used in this experiment, which were donated by the Veterinary Pathology Laboratory of the College of Veterinary Medicine, Shanxi Agricultural University.

### 2.3. Reagents

PrimeSTAR^®^ (Ōsaka shi, Japan) Max DNA Polymerase, DL2000 DNA Marker, DL5000 DNA Marker, pMD™ (Ōsaka shi, Japan) 18-T Vector Cloning Kit, *E.coli* BL21(DE3) Competent Cells, *E.coli* DH5α Competent Cells, QuickCut™ (Ōsaka shi, Japan) EcoR I, QuickCut™ Xho I, T4 DNA Ligase, TritonX-100, Tris, Glycine, SDS, Acrylamide, and SYBR Green qPCR Kit were purchased from TaKaRa (Ōsaka shi, Japan); pET-28a (+) DNA was purchased from Sangon Biotech (Shanghai, China); isopropyl β-D-thiogalactoside (IPTG), LB broth powder, LB agar powder, Ampicillin, and Kanamycin were purchased from Solarbio (Beijing, China); 6*His, His-Tag Monoclonal antibody, HRP-conjugated Affinipure Goat Anti-Rabbit IgG(H+L), HRP-conjugated Affinipure Goat Anti-Mouse IgG(H+L), Fluorescein (FITC)-conjugated Affinipure Goat Anti-Rabbit IgG(H+L), and ECL detection kit were purchased from Proteintech (Chicago, IL, USA).

### 2.4. Plasmid Construction

Specific primers were designed to measure the ROP27 sequence (Cluster-10347.5943) based on the transcriptome, and they were sent to a company for synthesis. Primer sequences were as follows: ROP27 sense primer, 5′-GAATTCAGGTAACGAGTCTCTGC-3′; and anti-sense primer, 5′-TTGCCAGAATTGGCTCTACTACG-3′. The PCR product was purified and cloned into the pMD18-T vector, according to the operation steps of the gel extraction kit. A single bacterial colony was selected for amplification, culture, and sequencing, and the bacterial solution with the most correct sequencing was selected for plasmid extraction. The plasmid and pET-28a (+) DNA were digested using EcoR I and Xho I endonucleases, and the digested products were purified. The target fragment was cloned into the eukaryotic expression vector, and then, a single colony was selected for amplification, culture, and sequencing. The plasmid with the correct sequencing was named pET-*Et*ROP27.

### 2.5. Protein Expression and Purification

The recombinant plasmid pET-*Et*ROP27 was transformed into BL21 (DE3) Competitive Cells. Subsequently, positive colonies were selected for enrichment. IPTG with a final concentration of 1 mmol/L was added to induce expression for 8 h when the OD_600_ value of the bacterial solution was about 0.5. The heavy suspension liquid was placed on ice for ultrasonic crushing treatment, and then, the sediment and supernatant were collected. We followed the method of protein purification as described by Barkhordari F. Protein purity and concentration were determined using 12% (*w*/*v*) sodium dodecyl sulfate polyacrylamide gel electrophoresis (SDS-PAGE) and the BCA protein determination kit. The purified protein was stored at −80 °C and used for subsequent experiments.

### 2.6. Preparation of Anti-E. tenella and Anti-rEtROP27 Positive Serum

In this experiment, 1 × 10^4^ sporulated *E. tenella* oocysts were intragastrically administered to 2-week-old SPF chickens. After 3 days, each chicken was infected with 5000 sporulated oocysts every 3 days for a total of four times. Blood was collected with cardiac puncture and centrifuged to prepare anti-*E. tenella* positive serum; this was then stored at −80 °C for standby.

Three 2-month-old New Zealand white rabbits were immunized with r*Et*ROP27, which was emulsified with Freund’s adjuvant at a dose of 200 µg per rabbit three times, once every two weeks. Before each immunization, 30 µL of rabbit blood was taken from the ear vein, and the serum was isolated to detect the antibody titer. Blood was collected from the heart and centrifuged to collect serum after the third immunization. The collected serum was deactivated in a water bath at 56 °C for 30 min and stored at −80 °C for standby.

### 2.7. Indirect ELISA Determination of Antibody Titer

Rabbit anti-*Et*ROP27 serum is the sample to be tested, and the non-immunized serum is the negative control. The purified *Et*ROP27 recombinant protein was used as the detection antibody. PBST was used to test the serum according to gradient dilutions of 1:500, 1:1000, 1:2000, 1:4000, 1:8000, 1:16,000, and 1:32,000. Serum antibody titer was tested according to the method reported by Dong-chao Zhang [18].

### 2.8. Western Blotting Analysis

The protein samples were separated with 12% polyacrylamide gels. After separation, the target band was transferred to 0.22 µm polyvinylidene fluoride membranes using the Bio-Rad wet transfer system. After blocking with 5% BSA in PBS at 37 °C for 1 h, the membranes were probed with anti-*E. tenella* (1:400), anti-r*Et*ROP27(1:500), and anti-HIS (1:2000), respectively, with an overnight incubation at 4 °C. Subsequently, the membranes were washed four times with PBS containing tween-20 (PBST) for 10 min each time. Next, the membranes were incubated with horseradish peroxidase-linked secondary anti-chicken, anti-rabbit, or anti-mouse IgG antibodies for 1 h at 37 °C. After washing four times with PBST for 10 min each time, the signal was visualized using an ECL hypersensitivity detection kit.

### 2.9. Primary Culture of Chicken Embryo Caecal Epithelial Cells

The 15-day-old SPF chicken embryo was placed in a bioclean bench. The cecum was isolated and thoroughly washed with the PBS buffer. Then, it was cut into a tissue of approximately 1 mm^3^ size. After washing, the tissue was resuspended and mixed with thermolysin (50 mg/L) and digested under shaking at 41 °C for 2 h; the PBS buffer was used to stop the digestion, and the supernatant was pipetted and centrifuged at 1200 r/min for 5 min. The PBS and enzyme solution were discarded; the cell pellet was resuspended in 10% fetal bovine serum (FBS) low-glycemic DMEM cell culture medium. The supernatant was collected after 70 min of adherent culture. After centrifugation at 1200 r/min for 5 min, it was resuspended and precipitated with DMEM/F12 culture medium containing 2.5% FBS. The cells were inoculated into a cell culture plate for cultivation. This culture can be used for subsequent experiments when the cell adhesion rate reaches 85%.

### 2.10. E. tenella Sporozoite Preparation

An appropriate amount of *E. tenella* sporulation oocysts was removed and centrifuged at 2000 r/min for 5 min, and the supernatant was discarded. The sediments were washed thoroughly with PBS. The oocysts were suspended in 2 mL PBS. This suspension was grinded with a homogenizer until its shelling rate reached 80%. The solution was centrifuged at 1800 r/min for 5 min. An appropriate amount of sporocyst digestion solution containing 0.75% trypsin and 10% chicken bile was added to the precipitate and was shaken for digestion at 41 °C (150 r/min) until 80% of the spores were released. After filtration, the solution was centrifuged at 3000 r/min for 10 min, and then suspended and precipitated in the DMEM culture medium. The amount of sporozoites used in this experiment was 1 × 10^4^.

### 2.11. RT-PCR Analysis

Total RNA from chicken cecum or primary cecal epithelial cells of chicken embryo cells infected with *E. tenella* precocious strains was extracted using TRIzol Reagent. The cDNA was generated with reverse transcription using PrimeScript reverse transcriptase. ROP27 mRNA was assessed using quantitative RT-PCR with a TaKaRa SYBR Green reagent. RT-PCR Kit was used with a CFX96 Touch real-time PCR detection system. The primer sequences were as follows: ROP27 sense primer, 5′-AGCTACGACACTCCTGTTGC-3′; and anti-sense primer, 5′-ACTCAAGACGGAGTTGCTGG-3′. The PCR product was purified with a gel extraction kit and then cloned into the pMD18-T vector. The extracted plasmid was continuously diluted and used as a standard for quantitative analysis. The initial copy number of ROP27 gene in each group was calculated using the following formula: X = −K log Ct + b, where X is the initial copy number, and K, Ct, and b refer to the slope rate, the cycle threshold, and a constant, respectively.

### 2.12. Animal Experiment

Firstly, 14-day-old chickens were randomly divided into 5 groups, with 10 chickens in each group. In three experimental groups, legs of the chicken were subcutaneously injected with recombinant proteins emulsified using Freund’s complete adjuvant (50 μg, 100 μg, and 150 μg, respectively). After 7 days, the recombinant protein was mixed with Freund’s incomplete adjuvant in equal volume and fully emulsified for enhanced immunity. The infection control group (challenged control) and the uninfected control group (unchallenged control) were injected with PBS and Freund’s complete adjuvant or Freund’s incomplete adjuvant at the same time. After 7 days, each chicken in the experimental group and the infection control group received an oral administration of 5 × 10^4^ sporulation oocysts of *E. tenella*, whereas PBS was administered to the uninfected control group (Table 1).

### 2.13. Concentration of Serum Antibody

The concentration of IgY antibody level in serum were detected using ELISA commercial kits, which were named as “chick cytokine ELISA Quantitation Kits” (catalog number: CSB-E11635Ch, for IgY CUSABIO, Wuhan, China), according to manufacturer’s instructions.

### 2.14. In Vivo Immunoprotective Parameters 

In vivo immunoprotective parameters of the r*Et*ROP27 protein include clinical, pathological, and parasitological factors [19]. Clinical factors include weight gain, survival, and mortality rate, which can be directly obtained and calculated. The pathological factor is the cecal injury score, which is a continuous number from 0 (none) to 4 (severe), representing varying degrees of cecal lesions, evaluated by three independent observers, according to Johnson and Reid [20]. Parasitological factors include oocyst output and ACI. The output of oocysts mainly detects the content of cecum per gram of oocysts (OPG), calculated using McMaster counting technique [21]. ACI is a synthetic criterion that can measure the effectiveness of anticoccidial activity. The calculation formula for ACI = (relative weight gain rate + survival rate) × 100 − Mean lesion score × 10 − OPG value). ACI > 180 indicates good protection, 160 < ACI < 179 indicates moderate protection, 120 < ACI < 159 indicates limited protection, and ACI < 120 indicates no protection [22].

### 2.15. Image and Statistical Analyses

All data were analyzed using SPSS 17.0 statistical software (Chicago, IL, USA) and expressed as arithmetic mean ± standard deviation. Histograms were prepared via GraphPad Prism 5.0 software (San Diego, CA, USA). Each experiment was repeated at least 3 times. The results are considered statistically significant when the *p*-value is less than 0.05.

## 3. Results

### 3.1. Gene Clone and Plasmid Construction

*Et*ROP27 showed a specific band at 1068 bp after PCR amplification. Gene sequence and bioinformatics were analyzed, details in Appendix A. The PCR product was then cloned into the pMD-18T vector, and positive colonies were selected for bacterial liquid PCR and sequencing analysis (Figure 1A,B). The target fragment was cut out from the T vector and cloned into the pET-28a vector. Then, the positive colonies were selected with bacterial liquid PCR for double digestion identification and sequencing analysis (Figure 1C,D). The prokaryotic expression vector pET-*Et*ROP27 was constructed successfully.

### 3.2. Expression and Purification of rEtROP27 Protein

Coomassie brilliant blue staining showed that the highest expression of *Et*ROP27 was observed after 6 h of IPTG induction (Figure 2A). After purification, the target band appears at about 48 kDa, and the elution buffer with a concentration of 150 mmol/L imidazole has the best elution effect (Figure 2B). Western blotting results showed that His tag anti body and chicken serum infected with coccidia were primary antibodies that could produce clear and reasonably sized target bands, while healthy chicken serum as primary antibody did not show any bands (Figure 2C). Chicken naturally infected with coccidia produced antibodies against *Et*ROP27. The results indicated that *Et*ROP27 was a natural antigen of *E. tenella* and could be a potential vaccine protein candidate.

### 3.3. Determination of rEtROP27 Polyclonal Antibody Titer

The rabbit anti-*Et*ROP27 serum was the sample chosen to be tested; the non-immunized serum was the negative control. The purified *Et*ROP27 recombinant protein was used as the detection antibody. After three immunizations with *Et*ROP27 fusion protein, blood and serum were collected to determine the antibody titer. From Table 2, it can be seen that the rabbit serum titer reaches 1:2.56 × 10^4^ after three immunizations. This indicates that the immunogen injection achieves a good immune effect, and the positive serum can be used for subsequent experiments. Western blotting results showed that r*Et*ROP27 protein could be detected with *Et*ROP27 polyclonal antibody (Figure 3).

### 3.4. Changes in EtROP27 Expression

The expression of *Et*ROP27 mRNA and protein was detected at different time points in primary chicken embryo cecal epithelial cells or 14-day-old chickens infected with *E. tenella*. The result showed that the expression of *Et*ROP27 mRNA in the primary cecal epithelial cells of chicken embryos infected with *E. tenella* gradually increased with the prolongation of infection time, reaching a peak at 96 h, and then decreasing. The protein expression pattern was also consistent with the one described above (Figure 4A–D). The results showed that *Et*ROP27 had the highest expression level during the first generation of schizogenesis.

### 3.5. In Vivo Immunoprotective Effect of rEtROP27 Protein 

Four chickens died in the challenged control group, while no deaths were observed in the other groups. Average and relative body weight gains of chickens that were immunized with the r*Et*ROP27 protein were significantly higher than in the challenged control groups (Figure 5A,B). Similarly, chickens vaccinated with the r*Et*ROP27 protein showed a reduction in the oocyst output compared with the challenged control group (Figure 5C,D), and lower lesion scores and bloody stool (refer Appendix A) compared with the infection control group. The ACI value of the low-dose group (50 μg) of the r*Et*ROP27 protein was more than 155, and that of the medium-dose group (100 μg) and the high-dose group (150 μg) was more than 160 (Table 3). The results indicated that the optimal immune dose for protein was 100 μg.

The IgY antibody levels of the serum with different doses of *Et*ROP27 in immune groups were significantly higher than those of the control group at 7 days after the second immunization (0 day post infection), with the group administered 100 μg of *Et*ROP27 having the highest antibody level (Figure 5E). The serum IgY antibody levels in the *Et*ROP27 immune group with different doses and in the challenged group at 7 day post infection were significantly higher than those at 0 day post infection (Figure 5E). These results indicated that *Et*ROP27 was an antigen of *E. tenella* and could induce a higher level of humoral immune response for resisting an *E. tenella* infection.

## 4. Discussion

Chicken coccidiosis is one of the most important diseases threatening the health and welfare of poultry [23,24,25]. There are poultry farming enterprises in various provinces of China, and the overall infection rate of chicken coccidia in China is over 50%, which seriously hinders the development of China’s poultry industry [3]. The life cycle of *E*. *tenella* is relatively complex and can be generally divided into two developmental stages: the exogenous stage (spore-stage reproduction) and the endogenous stage (schizogeny and gametogenesis). Coccidial infection impairs the nutrient uptake by its host, and the formation and release of merozoites and gametophytes destroy cell structures and functions, causing hemorrhagic inflammation, emaciation, and death in the cecum during *E. tenella* schizogeny and gametogenesis [26]. At present, the control of coccidiosis mainly relies on anticoccidial drugs, but they have many shortcomings. Therefore, there is an urgent need to develop a safe and effective vaccine against chicken coccidia to effectively prevent and control this disease.

*E. tenella* is a member of the apicomplexan protozoa [27]. The protozoa of the *apicomplexan* have unique secretory organelles, including rod-shaped bodies, dense particles, and microlines, which allow invasion mechanisms against host cells [28]. The secretion of rhoptry neck protein (RON) related to parasite invasion in the early stage is crucial for the formation and function of mobile connections between parasites and host cell membranes, such as RON1, RON2, and RON3 [29]. The release of ROPs from the base of the rod-shaped body into the PV changes the environment of the PV and enters the host cell [30,31]. They interact with the host cell through signal transduction, causing a series of downstream effects and becoming a key determinant of protozoan virulence. After entering the cell, ROPs can disrupt host cell signaling and defense mechanisms and assist in recruiting host organelles [32].

*Et*ROP plays an important role in the parasitic invasion of host cells, the modification of host vacuoles, and the regulation of host cells, and it has been proven to be a key virulence factor. It also has the properties of a protective antigen, and its corresponding antibodies can inhibit the infection of the host by coccidia [32]. Studies have confirmed the expression of *Et*ROP17 in *E. tenella* merozoites. Western blotting analysis showed that *Et*ROP17 can be recognized by the chicken immune system and induce antibody responses. The vaccination of animals with r*Et*ROP17 can significantly reduce cecal lesions [33]. This indicates that *Et*ROP17 can be used as an effective vaccine candidate for *E. tenella*. Similarly, studies have shown that *Et*ROP30 and *Et*ROP35 are natural antigens of coccidia, which have a good immunogenicity and are potential vaccine protein candidates [34,35].

*Et*ROP27 is a differential gene detected by the transcriptome, which is presumed to be a pathogenic gene [16,17]. However, its expression and function are still unclear. The different expression levels of secreted proteins at different stages of the parasite’s life cycle are often closely related to their function. A high expression of proteins in the mitotic stage of invading cells is associated with parasite development and host–parasite interactions [36]. The extracellular stages (sporozoite and merozoite) of *Eimeria* are fragile, but it is crucial for the invasion and development of the parasite [37]. The high expression of proteins in the sporozoite and/or merozoite stage makes *Et*ROP27 an ideal vaccine candidate. In this study, we first conducted a bioinformatics analysis on *Et*ROP27. The analysis results indicate that the protein is a secreted protein with potential antigenic properties. Subsequently, a prokaryotic expression recombinant plasmid for *Et*ROP27 was successfully constructed and purified, and polyclonal antibodies against *Et*ROP27 were successfully obtained. RT-PCR and Western blotting results indicate that *Et*ROP27 protein is naturally present and expressed at all stages of *E. tenella* development, with the highest expression level in the first generation schizonts. Therefore, it may be a good protein candidate for vaccines.

Subunit vaccines are expected to become an alternative control strategy for avoiding shortcomings of anticoccidial medication and live vaccines, and the screening of effective immunoprotective proteins is a key focus of the subunit vaccine research. Animal experiments showed that the r*Et*ROP27 protein could produce increased average body weight gain, increased IgY titer in serum, decreased oocyst output and bloody stool, and lower mean lesion scores compared with the infection control. The ACI value of r*Et*ROP27 protein for a dose of 100 μg or 150 μg reached more than 160. These results indicate that *Et*ROP27 can effectively induce immune protection against *E. tenella* in chickens. In this study, other doses, immune methods, immune ages, immune intervals, and other factors that may affect the final immune protection effect were not considered. Therefore, the immune program used in this study may not be optimal. A large number of scientific studies are still needed before the successful clinical application of r*Et*ROP27.

This study provides a basis for determining the key genes that dominate or regulate the pathogenicity of coccidia, for deciphering drug action targets that control the pathogenicity of coccidia. This study also provides new ideas and scientific basis for further revealing the pathogenic mechanism of coccidia against its host.

## 5. Conclusions

This study successfully constructed the eukaryotic expression vector of *E. tenella* ROP27 and purified its recombinant protein. Our findings suggest that ROP27 is a natural protein of *E. tenella* and is highly expressed during the first generation of schizogenesis. The results of animal experiments showed that medium-to-high doses of r*Et*ROP27 could have a moderate effect against coccidia, indicating that this gene can reduce the effects of *E. tenella* infection and providing new ideas for the prevention of *E. tenella* disease and the development of vaccines for the disease.

## Figures and Tables

**Figure 1 animals-13-03500-f001:**
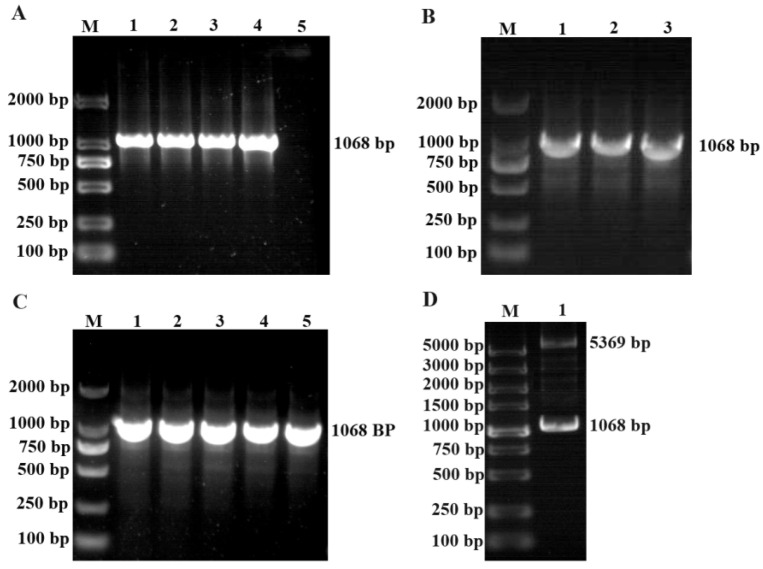
The construction of pET-*Et*ROP27 Expression Vector. (**A**) *EtROP27* gene amplification. 1–4: annealing temperatures are 52 °C, 54 °C, 56 °C, and 58 °C, respectively. 5: negative control. (**B**) The bacterial solution PCR results of pMD-*Et*ROP27. 1–3: monoclonal colony amplified bacterial liquid. (**C**) The bacterial solution PCR results of pET-*Et*ROP27. (**D**) The results of double enzyme digestion of pET-*Et*ROP27. M: DNA marker.

**Figure 2 animals-13-03500-f002:**
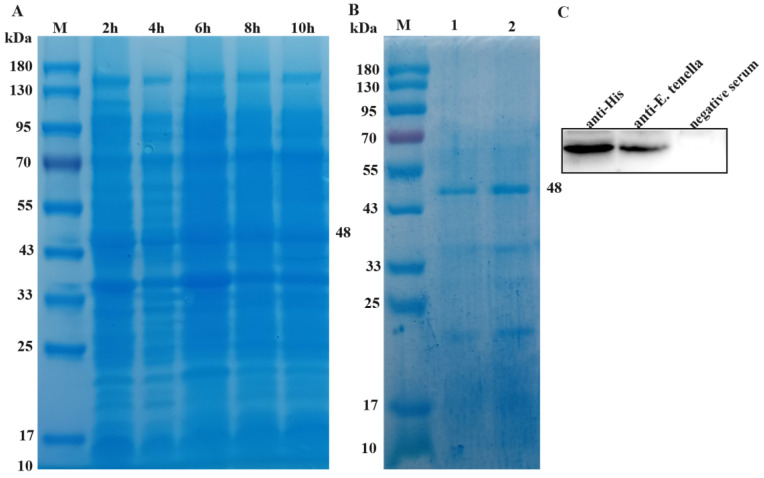
The expression, purification, and identification of r*Et*ROP27. (**A**) Coomassie bright blue staining of rEtROP27 protein. (**B**) Coomassie brilliant blue staining of purified r*Et*ROP27 protein. 1: 100 mmol/L imidazole. 2: 150 mmol/L imidazole. (**C**) Western blotting identification results.

**Figure 3 animals-13-03500-f003:**
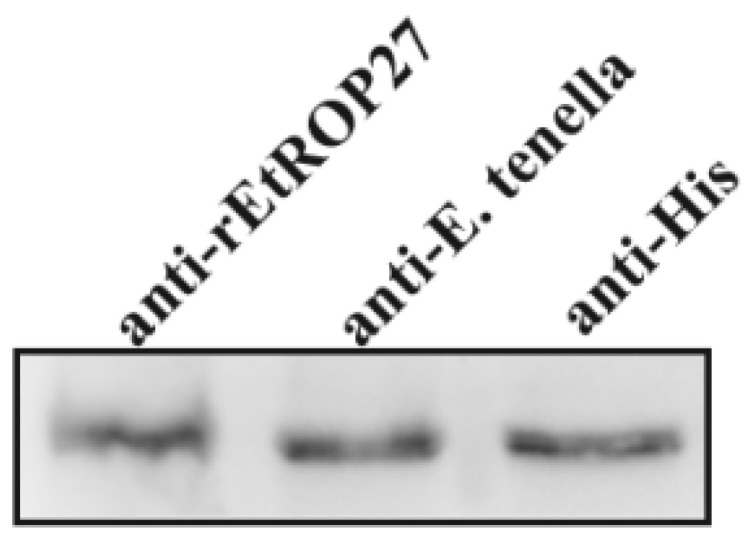
Specific detection of *Et*ROP27 polyclonal antibody.

**Figure 4 animals-13-03500-f004:**
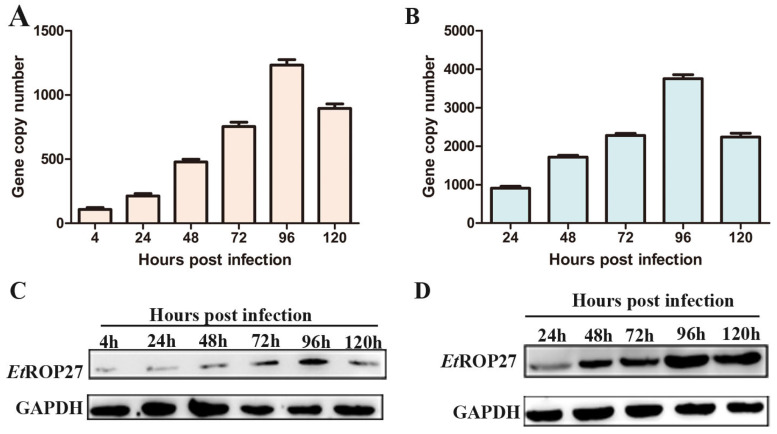
The expression of *Et*ROP27 in cells and tissues. (**A**,**B**) The expression of *Et*ROP27 mRNA was detected using qRT-PCR. A: chicken embryo cecal epithelial cells. (**B**) cecal tissue. (**C**,**D**) The expression of *Et*ROP27 protein was detected using Western blotting. (**C**) chicken embryo cecal epithelial cells. (**D**) cecal tissue. All of the data are representative of at least three independent experiments.

**Figure 5 animals-13-03500-f005:**
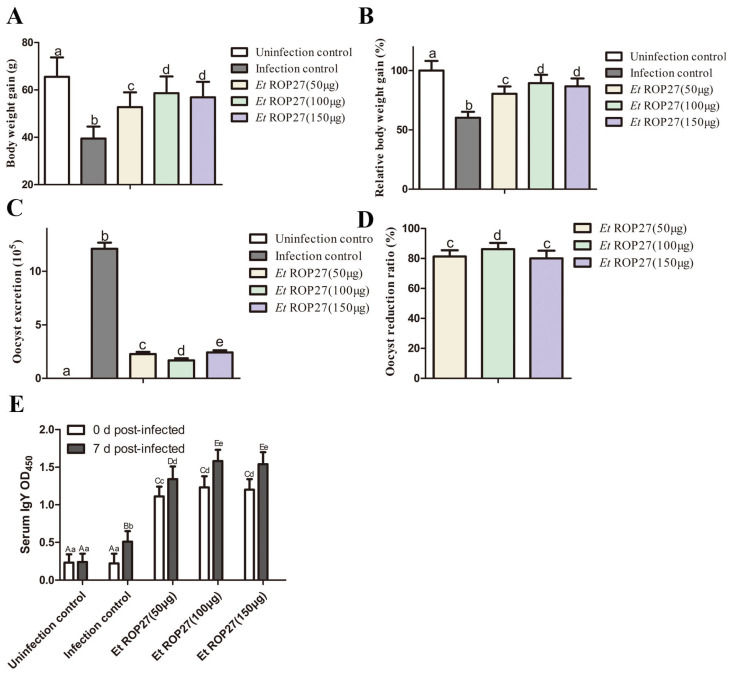
Effects of recombinant *Et*ROP27 protein against the *E. tenella* challenge. (**A**) Body weight gain. (**B**) Relative body weight gain. (**C**) OPG. (**D**) Oocyst reduction ratio. (**E**) IgY concentration. All of the data are representative of at least three independent experiments. The same column of shoulder markers with the same lowercase letters indicates no significant difference (*p* > 0.05), shoulder markers with different lowercase letters indicate significant differences (*p* < 0.05), shoulder markers with different uppercase letters indicate extremely significant differences (*p* < 0.01).

**Table 1 animals-13-03500-t001:** Grouping of experimental animals and immune challenge procedures.

Groups	Immunization	Dose	Challenge
Unchallenged control	PBS + adjuvant	/	/
Challenged control	PBS + adjuvant	/	*E. tenella* sporulated oocysts (5 × 10^4^)
*Et*ROP27 (50 μg)	r*Et*ROP27 protein + adjuvant	50 μg	*E. tenella* sporulated oocysts (5 × 10^4^)
*Et*ROP27 (100 μg)	r*Et*ROP27 protein + adjuvant	100 μg	*E. tenella* sporulated oocysts (5 × 10^4^)
*Et*ROP27 (150 μg)	r*Et*ROP27 protein + adjuvant	150 μg	*E. tenella* sporulated oocysts (5 × 10^4^)

**Table 2 animals-13-03500-t002:** Titer determination of *Et*ROP27 polyclonal antibody (OD_450_).

Antibody Dilution Ratio									Negative Serum
1:100	1:200	1:400	1:800	1:1600	1:3200	1:6400	1:12,800	1:25,600	
*	*	*	3.442	2.532	2.015	1.503	1.022	0.516	0.36

Note: * indicates that the OD_450_ value exceeds 3.5.

**Table 3 animals-13-03500-t003:** Experimental ACI value for chicken.

Group	Relative Body Weight Gain (%)	Oocyst Index	Cecum Mean Lesion Score	Anticoccidial Index
Unchallenged control	100	0	0	200.00
Challenged control	60.25	5	2.9	87.25
*Et*ROP27 (50 μg)	80.45	1	2.0	159.45
*Et*ROP27 (100 μg)	89.47	1	1.7	171.47
*Et*ROP27 (150 μg)	86.75	1	1.9	166.75

## Data Availability

Data are contained within the article and Appendix A.

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
