# Peer review of "The Immunoprotective Effect of ROP27 Protein of Eimeria tenella"

_animals, 2023, doi:10.3390/ani13223500_

Round 1

Reviewer 1 Report

Comments and Suggestions for Authors

The manuscript entitled as “The immunoprotective analysis of ROP 27 protein in Eimeria tenella.” is a nice, well planned and need of present era in which authors discussed the role of EtROP27 in immunogenicity of Eimeria tenella. However, I have some comments to improve the quality of this manuscript.

-please write the scope of the Poultry industry in China in the start sentence of the introduction and discussion.

-The English language should be improved before publication.

-results section. Please move some supportive results to the supplementary file.

-Please write “Fluorescence intensity was analyzed using ImageJ software (National Institutes of Health, USA)” in the respective section. Not in statistical analysis.

- did the data was normalized before statistical analyses.

-did the data was employed posthoc test after ANOVA.

- how you have selected the number of chickens statistically.

-results- Figure 1 is not so much readable.

-figure 5- please make the graph colored.

-please some more relevant and recent references because the number of references are too less.

Comments on the Quality of English Language

The English language should be improved before publication.

Reviewer 2 Report

Comments and Suggestions for Authors

This is very interesting paper. Experiments were well designed and obtained data clearly showed that EtROP27 is a promising candidate for a vaccine against Eimeria tenella-caused avian coccidiosis. 

    However, there are several flaws that should be addressed before acceptance of this paper. Comments are as follows. 

Line 45, I don’t understand the meaning of “blood, stool, or even blood”. Does it mean “Bloody stool, or even blood only”? Please consider.

Lines 190-198, These sentences are very difficult to understand for me. I don’t know why adjuvant and immunization methods are different between experimental (challenge) and controls? Did you use complete Freund's adjuvant for all 5 groups including experiment and controls at the first immunization? And incomplete Freund's adjuvant for the second immunization? Please consider that and if necessary, you should modify the sentence adequately. 

Line 285, There is double “in”, so one should be removed.

Lines 315-340, The first and second sentences of the “Discussion” are almost repetition of parts of “Introduction”. That should be removed are very shortened with necessary discussions. You clearly showed that ROP27 immunization induced oocyst output, caecum injuries, reducing body weight loss. I think these results should be emphasized and discussed at first. 

Immunization methods and antigen dose should be improved before applicable use of EtROP27, so this point also be discussed.

Reviewer 3 Report

Comments and Suggestions for Authors

Dear Authors,

The paper submitted for evaluation is very interesting and contains interesting experimental results. It was written in clear language, easy for the reader to understand. The study concerns an important issue: the good health condition of chickens. I believe that the research carried out is preliminary and therefore it is justified to continue it. Apart from minor editorial comments, I have no other objections. The paper should be published in the current form.

Minor comments:

- Figure 1 is illegible, I suggest the authors consider a different way of presenting this data

- similarly, in Figure 6 the arrows are not clearly visible

- are further studies being conducted? will they continue?

Best regards,
